# Requirement of Zebrafish Adcy3a and Adcy5 in Melanosome Dispersion and Melanocyte Stripe Formation

**DOI:** 10.3390/ijms232214182

**Published:** 2022-11-16

**Authors:** Lin Zhang, Meng Wan, Ramila Tohti, Daqing Jin, Tao P. Zhong

**Affiliations:** Shanghai Key Laboratory of Regulatory Biology, Institute of Molecular Medicine, School of Life Sciences, East China Normal University, Shanghai 200241, China

**Keywords:** Adcy3a, Adcy5, melanosome dispersion, melanocyte stripe formation

## Abstract

cAMP-PKA signaling plays a pivotal role in melanin synthesis and melanosome transport by responding to the binding of the α-melanocyte-stimulating hormone (α-MSH) to melanocortin-1 receptor (MC1R). Adenylate cyclases (ADCYs) are the enzymes responsible for the synthesis of cAMP from ATP, which comprises nine transmembrane isoforms (ADCYs 1-9) and one soluble adenylate cyclase (ADCY 10) in mammals. However, little is known about which and how ADCY isoforms regulate melanocyte generation, melanin biosynthesis, and melanosome transport in vivo. In this study, we have generated a series of single and double mutants of Adcy isoforms in zebrafish. Among them, *adcy3a^-/-^* and *adcy5^-/-^* double mutants cause defects in melanosome dispersion but do not impair melanoblast differentiation and melanocyte regeneration during the embryonic or larval stages. Activation of PKA, the main effector of cAMP signaling, significantly ameliorates the defects in melanosome dispersion in *adcy3a^-/-^* and *adcy5^-/-^* double mutants. Mechanistically, Adcy3a and Adcy5 regulate melanosome dispersion by activating kinesin-1 while inhibiting cytoplasmic dynein-1. In adult zebrafish, Adcy3a and Adcy5 participate in the regulation of the expression of microphthalmia transcription factor (Mitfa) and melanin synthesis enzymes Tyr, Dct, and Trp1b. The deletion of Adcy3a and Adcy5 inhibits melanin production and reduces pigmented melanocyte numbers, causing a defect in establishing adult melanocyte stripes. Hence, our studies demonstrate that Adcy3a and Adcy5 play essential but redundant functions in mediating α-MSH-MC1R/cAMP-PKA signaling for regulating melanin synthesis and melanosome dispersion.

## 1. Introduction

Melanocytes are specialized cells that produce melanin within unique organelles called melanosomes. Melanin synthesis and pigmentation not only determine the colors of hair and skin but also are a critical protective mechanism for the skin against ultraviolet radiation, which is a major risk factor for melanoma and skin cancers. In the melanin biosynthesis pathway, α-MSH binds to MC1R, a G-protein-coupled receptor (GPCR), and activates ADCYs to promote cyclic AMP (cAMP) production. The increased intracellular cAMP activates protein kinase A (PKA), which phosphorylates cAMP-response element-binding protein (CREB), leading to the expression of MITF [1,2]. MITF is a major transcription factor for melanogenesis, which promotes the expression of melanogenic enzymes, including tyrosinase (TYR), tyrosinase-related protein 1 (TRP1), and tyrosinase-related protein-2 (TRP2/DCT), thereby increasing eumelanin synthesis [1,2,3]. Both *mitf^-^^/^^-^* null mutant mice and zebrafish *nacre/mitfa* mutants almost completely lack melanocytes [4,5].

After melanin is synthesized, mature melanosomes are transported along microtubule and actin filaments by employing microtubule-based motors and actin-dependent motors [6,7]. Microtubule motors move in opposite directions, with plus-end-directed kinesin motor proteins dispersing melanosomes towards the cell periphery (anterograde), whereas minus-end-directed cytoplasmic dynein-1 motor is responsible for melanosome aggregation to the perinuclear region (retrograde) [6,7]. In mammals, cAMP signaling is involved in melanocyte dendrite formation, and promotes melanosome peripheral transport as well as accumulation at the dendrite tips of melanocytes [8,9]. In fish and amphibian melanocytes, α-MSH-MC1R/cAMP-PKA signaling is critical for melanosome transport [7,10,11,12]. Administration of exogenous α-MSH causes a significant dispersion of melanosomes [7,13]. The activation of ADCYs by forskolin enhances an accumulation of intracellular cAMP, leading to melanosome dispersion [10,11,12]. In contrast, genetic knockdown of melanocortin 1 receptor (MC1R) or pharmacological inhibition of PKA prevents melanosome dispersion and contributes to melanosome aggregation [10,13].

In mammals, the ADCY family consists of nine transmembrane isoforms (tmAC) and one soluble isoform (sAC). cAMP-mediated signaling pathways play critical roles in melanocyte survival, proliferation, pigment synthesis, as well as melanoma progression [1,3]. Although in vitro studies have shown that knockdown of Adcy5, Adcy6, and Adcy9 isoforms, respectively, appears to reduce a-MSH-stimulated melanogenesis in mouse B16F10 melanoma cells [14], none of the individual Adcy isoform knockout mice show defects in the pigmentation of skin and hair. Therefore, which and how Adcys regulate melanocyte generation, melanin synthesis, and melanosome transport in vivo remains unknown.

Zebrafish have become a valuable model for studying pigment pattern formation and human pigment cell disease due to the visible process of melanocyte development and conserved melanogenesis pathways. In the present study, we generated a series of zebrafish *adcy* mutants, including *adcy3a*^-/-^, *adcy5*^-/-^, and *adcy6a*^-/-^ single mutants, as well as double mutants *adcy3a*^-/-^;*adcy5*^-/-^ and *adcy6a*^-/-^;*adcy6b*^-/-^. By dissecting melanosome distribution in melanocytes among these mutants, we demonstrate that Adcy3a and Adcy5 are the critical regulators in guiding melanosome transport by acting downstream PKA and modulating the activities of microtubule motor kinesin-1 and cytoplasmic dynein-1. In addition, we reveal that *adcy3a* and *adcy5* double mutations interfere with melanin synthesis by downregulating the gene expression of *mitfa* and melanogenic enzymes *tyr*, *trp1b,* and *dct*, leading to a failure to establish adult melanocyte stripes. Overall, our findings reveal novel roles of Adcy3a and Adcy5 in regulating pigmentation in vertebrates and provide an optimal animal model for studying the roles of cAMP signaling in melanocyte development and potential skin malignancies such as melanoma.

## 2. Results

### 2.1. Generation and Analyses of Zebrafish adcy Mutants

To identify the key Adcy isoforms in regulating melanogenesis and melanosome transport, we used zebrafish as a model organism and generated several *adcy* mutants, including *adcy3a^-/^^-^, adcy5^-/^^-^,* and *adcy6a^-/^^-^,* using the CRISPR/Cas9 gene editing technology. These Adcy isoforms were previously reported to be relatively highly expressed in sorted melanocytes from zebrafish embryos [15]. The single guide RNA (sgRNA) was designed to target exon 6 of *adcy3a* and *adcy5* as well as exon 2 of *adcy6a,* respectively. Sequencing analyses identified two *adcy3a* mutations with a 7-base pair (bp) and a 17-bp deletion, two *adcy5* mutations with a 7-bp deletion and an 11-bp insertion, and a 7-bp deletion in *adcy6a* mutation (Figure 1A–C). Each of these Adcy mutations was predicted to produce a premature stop codon, which generates a truncated protein lacking an intact adenylate cyclase catalytic domain (Figure 1A–C). Since the two mutant lines in *adcy3a* or *adcy5* exhibited the same phenotypes, we used 7-bp deletion in the *adcy3a^-/^^-^* line and 7-bp deletion in the *adcy5^-/^^-^* line for further experiments in this study.

Zebrafish aggregate melanosomes in the melanocytes under bright backgrounds, while rapidly dispersing them under dark backgrounds [13,16]. This background adaptation of light responses is controlled by the cAMP signaling via two hormones, α-MSH and the melanin-concentrating hormone (MCH) [16,17]. α-MSH stimulates melanosome dispersion by increasing intracellular cAMP levels, while MCH induces melanosome aggregation by decreasing it [16,18]. Therefore, we took advantage of the background adaption response to assess the effects of each Adcy mutation on zebrafish embryo pigmentation. Wild-type (WT) embryos raised under a dark background until 4th-day post fertilization (dpf) displayed a complete dispersion of melanosomes on the head (Figure 1D), whereas *adcy5^-/^^-^* mutants but not *adcy3a^-/^^-^* or *adcy6a^-/^^-^* mutants presented aggregated melanosomes compared with WT embryos (Figure 1D–G). We selected a defined region of interest (ROI) on the head to measure the pigmentation area and found that the pigmentation coverage of ROI was significantly reduced in *adcy5^-/^^-^* mutants compared with that of WT embryos, which was consistent with the phenotype of melanosome aggregation in *adcy5^-/^^-^* mutants (Figure 1H). Nonetheless, the number of melanocytes of ROI had no significant differences between WT embryos and these *adcy* mutants (Figure 1I).

Zebrafish embryonic melanocytes arise directly from the neural crest and establish the embryonic population of melanocytes in a Mitfa-dependent manner during the first three days of development [19]. The *mitfa* gene, a marker gene for early melanoblast, is directly regulated by the neural crest transcription factor Sox10, which is necessary for the specification of all pigment cell lineages, including melanocytes, xanthophores, and iridophores [20,21,22]. To investigate whether *adcy5* mutation affected melanoblast specialization, we examined the expression of *sox10* and *mitfa*. Whole-mount in situ hybridization (WISH) analyses showed that their expression levels in *adcy5*^-/-^ mutants were comparable with those in WT embryos (Figure 1J–M). Meanwhile, *tyr* and *dct*, two differentiated melanocyte markers, also had similar expression levels between *adcy5*^-/-^ mutants and WT embryos (Figure 1N–Q). Together, these findings suggest that *adcy5* is required for regulating melanosome dispersion, but not melanoblast differentiation and melanocyte generation during embryonic development.

### 2.2. adcy5 and adcy3a Play Redundant Functions in Regulating Melanosome Dispersion

Zebrafish genome contains two *adcy3* paralogs, *adcy3a* and *adcy3b*, as well as two *adcy6* genes, *adcy6a* and *adcy6b.* Although RNA-seq data generated from sorted embryonic melanocytes in zebrafish show low expression levels of *adcy3b* and *adcy6b* [15], we still generated two double mutants, *adcy3a^-/^^-^*;*adcy3b^-/^^-^* and *adcy6a^-/^^-^*;*adcy6b^-/^^-^,* by knocking out *adcy3b* in *adcy3a^-/^^-^* as well as *adcy6b* in *adcy6a^-/^^-^* (Figure 1A,C; Appendix A). Unfortunately, both *adcy3a^-/^^-^*;*adcy3b^-/^^-^* and *adcy6a^-/^^-^*;*adcy6b^-/^^-^* double mutants exhibited normal melanosome distribution and melanocyte number as observed in WT embryos (Figure 2A–C,I,J). To further assess whether *adcy3a*, *adcy5*, and *adcy6a* have redundant functions in regulating pigmentation, we analyzed the phenotypes of pairwise deficiencies among the three Adcys. The knockdown of *adcy6a* in *adcy3a^-/^^-^* mutants by injection of antisense morpholino oligonucleotides (MOs) had no significant effects on pigmentation as seen in WT embryos (Figure 2A,D,I). Similarly, *adcy6a*-MO-injected *adcy5^-/^^-^* mutant embryos also fail to aggravate melanosome aggregation when compared to *adcy5^-/^^-^* mutants (Figure 2E,F,I). Conversely, a*dcy3a^-/^^-^*;*adcy5^-/^^-^* double mutants generated by crossing homozygous *adcy3a^-/^^-^* and *adcy5^-/^^-^* mutant lines showed a higher aggregation of melanosomes than *adcy5^-/^^-^* single mutants (Figure 2E,G). The pigment coverage in the ROI also dramatically decreased in *adcy3a^-/^^-^*;*adcy5^-/^^-^* double mutants compared with *adcy5^-/^^-^* single mutants (Figure 2I). Co-injection of *adcy3a* and *adcy5* messenger RNA could rescue the defect phenotype of melanosome aggregation in *adcy3a^-/^^-^*;*adcy5^-/^^-^* double mutants, demonstrating that *adcy3a* and *adcy5* are indeed the genes affected in *adcy3a^-/^^-^*;*adcy5^-/^^-^* double mutants (Figure 2G–I). Furthermore, we quantified the number of melanocytes in *adcy3a^-/^^-^*;*adcy5^-/^^-^* double mutants and found no significant alterations compared to WT embryos (Figure 2J). WISH analyses revealed that the expression levels of *sox10, mitfa, tyr*, and *dct* in *adcy3a^-/^^-^*;*adcy5^-/^^-^* double mutants were comparable with those in WT embryos (Figure 2K–R). Collectively, these findings indicate that Adcy3a participates in the regulation of melanosome dispersion dominated by Adcy5, while neither of them is involved in the differentiation and specialization of melanocytes during embryonic stages.

### 2.3. adcy5 and adcy3a Are Required for Melanin Patterning in Metamorphic and Adult Stages

To better understand the role of these Adcy isoforms in pigment pattern formation, we examined the formation of melanocyte stripes in these mutants at later metamorphosis (28–30 dpf). Similar to WT zebrafish, *adcy3a^-/^^-^* and *adcy6a^-/^^-^* single mutants, as well as *adcy3a^-/^^-^*;*adcy3b^-/^^-^* and *adcy6a^-/^^-^*;*adcy6b^-/^^-^* double mutants developed normal metamorphic melanocyte stripes, consisting of a dorsal (1D) and ventral primary melanocyte stripes (1V), where melanosomes present corolla-like distribution in melanocytes due to normal dispersion [23,24] (Figure 3A,A′; Appendix A). In *adcy5^-/^^-^* mutants, the metamorphic melanocyte stripes had been established but displayed a mixture of punctate and corolla-like patterns on melanosomes (Figure 3B,B′). Importantly, *adcy3a^-/^^-^*;*adcy5^-/^^-^* double mutants failed to establish visible metamorphic melanocyte stripes as WT zebrafish, and the corolla-like melanosomes were completely replaced by the punctate pattern of melanosomes (Figure 3C,C′), suggesting that the dispersion of melanosomes in these melanocytes were compromised due to *adcy5* and *adcy3a* double mutations. However, the number of melanocytes seemed not to be significantly reduced in the *adcy3a^-/^^-^*;*adcy5^-/^^-^* double mutants.

Next, we examined pigment pattern formation in these *adcy* mutants at 6-month-old adult stages. *adcy3a^-/^^-^* and *adcy6a^-/^^-^* single mutants, as well as *adcy6a^-/^^-^*;*adcy6b^-/^^-^* double mutants, exhibited normal adult body stripe patterns as seen in WT adult zebrafish, including three ventral black stripes 1V, 2V, and 3V, and two dorsal stripes, 1D and 2D, along the dorsoventral axis [25] (Figure 3D,D′,D′′; Appendix A). In contrast, *adcy5^-/^^-^* mutants only developed three black stripes, including 1D, 1V, and 2V (Figure 3E,E′,E′′). The density of pigmented melanocytes in the three black stripes appeared reduced, and the melanin granules became smaller, making the skin look pale (Figure 3E′,E′′). Remarkably, in *adcy3a^-/^^-^*;*adcy5^-/^^-^* double mutants, the melanocyte stripes almost completely disappeared, and only a few pigmented punctate melanocytes were scattered on the surface of the body skin (Figure 3F,F′,F′′). Visualization of the pigment stripes under a high-power microscope showed that many unpigmented melanocytes and a few incompletely pigmented melanocytes surrounded the pigmented melanocytes (Figure 3G,G′), suggesting that *adcy5* and *adcy3a* double mutations may not cause a reduction in the number of melanocytes, but rather suppress melanin synthesis in these unpigmented melanocytes. To confirm this, we examined the expression of *mitfa* and three melanin synthesis genes, *dct*, *tyr*, and *trp1b* in isolated adult mutant skin. Quantitative real-time PCR analyses revealed that the expression levels of these genes in *adcy3a^-/^^-^*;*adcy5^-/^^-^* mutants were significantly reduced compared to WT adult zebrafish (Figure 3H), which was consistent with the defective phenotype in pigmentation in *adcy3a^-/^^-^*;*adcy5^-/^^-^* mutants. Next, we examined the pigment pattern in the caudal, anal, and dorsal fins in *adcy3a^-/^^-^*;*adcy5^-/^^-^* double mutants and found that the black stripes of mutant fins were mainly composed of highly aggregated melanosomes, which is different from those of WT fins presenting visibly dispersed melanosomes (Figure 3I–N). Taken together, these findings suggest that *adcy3a* and *adcy5* double mutations disrupted melanin synthesis by repressing the expression of *mitfa* and melanogenic enzyme gene, ultimately resulting in the lack of mature melanocyte stripes in mutant adult zebrafish.

### 2.4. adcy5 and adcy3a Double Mutations Have No Effects on the Establishment of Melanocyte Stem Cells

In zebrafish, the embryonic pigment patterns are established during the first three days of development and gradually replaced by the adult pigment patterns during metamorphosis [26]. Melanocyte stem cells (MSCs) that reside at the dorsal route ganglia (DRG) are generated on the first day of development and are responsible for generating the adult population of melanocytes [19,27]. Defects in the establishment of MSCs in zebrafish pigment mutants, such as *picasso/erbb3b* and *sparse/kita*, caused a marked decrease in the number of adult stripe melanocytes [23,27,28,29]. Since *adcy3a* and *adcy5* double mutations did not affect the generation of adult melanocytes, except for deficient melanization of melanocytes, we reasoned that the establishment of MSCs might be normal. To test this idea, we carried out larval melanocyte regeneration experiments. Given that the DRG-associated MSCs are quiescent stem cells that can be recruited and differentiated into new melanocytes when melanocyte death [29,30], we ablated ontogenetic melanocytes using 2-morpholinobutyl-4-thiophenol (MoTP), a kind of melanotoxic chemical that can specifically ablate differentiated melanocytes that express tyrosinase [31]. Following the treatment of embryos with MoTP from 1 to 3 dpf, the differentiated melanocytes were almost completely ablated (Appendix A), we washed out MoTP and continued to raise the embryos in the normal medium until 6 dpf. Quantitative analysis of melanocyte number showed that there was no difference between *adcy3a^-/^^-^*;*adcy5^-/^^-^* mutants and WT larvae (Appendix A), suggesting that the formation of MSCs was unaffected by *adcy3a* and *adcy5* double mutations. Since Erbb3b signaling and Kita signaling are involved in MSC fate decision in zebrafish [28,29], we assessed the expression levels of two receptor tyrosine kinases, *erbb3b* and *kita,* in *adcy3a^-/^^-^*;*adcy5^-/^^-^* mutants by qPCR analyses, and no significant differences were detected between WT embryos and mutants (Appendix A), indicating that MSC fate determination is well established in *adcy3a^-/^^-^*;*adcy5^-/^^-^* mutants.

### 2.5. PKA Activation Rescues the Defects of Melanosome Dispersion in adcy3a^-/-^;adcy5^-/-^ Double Mutants

Previous studies have reported that cAMP-mediated PKA activation plays a critical role in the regulation of melanosome transport [6,32]. An increase in cAMP levels stimulates PKA activity and induces melanosome dispersion, whereas a decrease in cAMP levels suppresses PKA activity and leads to melanosome aggregation [6,7]. To determine whether activation of PKA signaling downstream of cAMP can rescue the defects of melanosome dispersion in *adcy3a^-/^^-^*;*adcy5^-/^^-^* double mutants, we treated 10-day-old mutants with dibutyl-cyclic AMP (db-cAMP), an analog of cAMP which specifically activates PKA. Remarkably, almost all melanocytes in *adcy3a^-/^^-^*;*adcy5^-/^^-^* double mutants had reacted to the treatment of db-cAMP within 30 min, exhibiting an increase in melanosome dispersion on the head, compared to the DMSO control group (Figure 4A–D,I). However, administration of exogenous α-MSH only induced mild melanosome dispersion in a few melanocytes in *adcy3a^-/^^-^*;*adcy5^-/^^-^* double mutants (Figure 4E,F,I), but caused a dramatic dispersion of melanosomes in all melanocytes in WT larvae (Figure 4G,H,J). Further, we evaluated whether the addition of exogenous α-MSH and db-cAMP can ameliorate melanosome aggregation in adult mutants. A piece of fresh caudal fin detached from *adcy3a^-/^^-^*;*adcy5^-/^^-^* mutants was exposed to α-MSH and db-cAMP for 30 min, respectively. In α-MSH-treated mutant caudal fins, melanosomes failed to disperse as in the DMSO control group (Figure 4K–N). Instead, almost all melanosomes in *adcy3a^-/^^-^*;*adcy5^-/^^-^* mutants caudal fin responded to db-cAMP treatments and dispersed noticeably within 30 min (Figure 4O,P). Collectively, these findings indicate that α-MSH signaling regulates melanosome dispersion in an Adcy3a/Adcy5-dependent manner, and PKA signaling acts downstream of Adcy3a/Adcy5 to regulate melanosome transport.

### 2.6. Activation of Kinesin-1 or Inhibition of Cytoplasmic Dynein-1 Facilitates Melanosome Dispersion in adcy3a^-/-^;adcy5^-/-^ Double Mutants

Melanosome dispersion along the microtubule is mediated by kinesin motors, while its aggregation depends on cytoplasmic dynein-1. In fish, high cAMP levels activate kinesin motors and suppress cytoplasmic dynein-1 during dispersion, whereas low cAMP levels reduce kinesin activity during aggregation [10,33]. We reasoned that *adcy3a* and *adcy5* double mutations might result in a reduction in cAMP levels, which inhibited kinesin motor activity. Increasing kinesin motor activity or reducing cytoplasmic dynein-1 function might ameliorate the phenotype of highly aggregated melanosomes in *adcy3a^-/^^-^*;*adcy5^-/^^-^* double mutants. To test this idea, we treated *adcy3a^-/^^-^*;*adcy5^-/^^-^* double mutants with 50 μM kinesore, an inducer of kinesin-1 activation [34]. As expected, kinesore-treated *adcy3a^-/^^-^*;*adcy5^-/^^-^* double mutants showed significant dispersion of melanosomes compared with DMSO controls (Figure 5F,G,L,M,M′,N,N′). Similarly, kinesore-treated wild-type embryos also exhibited high dispersion of melanosomes due to kinesin-1 activation (Figure 5A,B,K). Next, we attempted to restrain the activity of cytoplasmic dynein-1 in *adcy3a^-/^^-^*;*adcy5^-/^^-^* double mutants with a small-molecule inhibitor, ciliobrevin D [35]. Treatment of *adcy3a^-/^^-^*;*adcy5^-/^^-^* double mutants with 20 μM ciliobrevin D induce significant melanosome dispersion compared with controls (Figure 5F,H,L,M,M′,O,O′), which was also seen observed in DMSO-treated WT embryos (Figure 5A,C,K).

*Dynein 1 heavy chain 1 (dync1h1)* encodes a heavy chain subunit of cytoplasmic dynein-1. Dync1h1 mutation disrupts dynein’s complex assembly and impairs the function of the cytoplasmic dynein-1 motor [36]. Loss of *dync1h* causes abnormal melanosome dispersion in zebrafish embryos [37,38,39]. To further clarify that the disruption of cytoplasmic dynein-1 function might alleviate the high aggregation of melanosomes in *adcy3a^-/^^-^*;*adcy5^-/^^-^* mutants, a genetic knockdown of *dync1h1* was performed in double mutants by injecting *dync1h1-*MO. The *dync1h1-*MO injected mutant embryos showed significant dispersion of melanosomes compared with controls (Figure 5I,J,L,P,P′), which was in agreement with the positive effects treated by pharmacological inhibition of dynein-1. Meanwhile, the injection of *dync1h1-*MO into WT embryos also caused significant melanosome dispersion as previously reported (Figure 5D,E,K). Taken together, these findings indicate that *adcy3a* and *adcy5* double mutations repress kinesin-1 motor activity and promote cytoplasmic dynein-1-mediated melanosome aggregation.

## 3. Discussion

In this study, we have presented evidence that Adcy5 and Adcy3a play redundant roles in regulating melanosome dispersion and melanin production in zebrafish. Based on our findings and previous studies, we propose a model in which α-MSH binds to Mc1r, and stimulates Adcy3a and Adcy5 to increase intracellular cAMP. The cAMP-dependant PKA activation guides melanosome dispersion by activating Kinesin-1 and suppressing cytoplasmic Dynein-1, as well as promoting the expression of *mitfa* and its downstream target genes *tyr*, *trp-1*, and *trp-2*, ultimately leading to melanin synthesis (Figure 5Q). To our knowledge, this is the first report of two Adcy isoforms that work together in regulating melanosome transport, melanin biosynthesis, and melanin pigmentation in vivo.

The bidirectional melanosome transport is driven by the opposite action of the kinesin motor proteins and the cytoplasmic dynein-1 and then switches to myosin-driven transport along actin filaments. Increasing cAMP levels promotes melanosome dispersion by activating the kinesin-related motor proteins and repressing cytoplasmic dynein-1 while reducing cAMP levels induces melanosome aggregation by suppressing the kinesin-related motor proteins [10,33]. The cAMP-dependent PKA pathway has been reported to potentially modulate the activity of kinesins and cytoplasmic dynein-1 via phosphorylation and dephosphorylation processes [11,32]. Previous studies have reported that the *adcy5* mutation in guppy appears as punctate melanocytes, and the melanosomes in mutant melanocytes are unable to disperse by forskolin and IBMX treatments, which may be due to the deficiency in melanocyte differentiation and/or growth, but not in melanosome dispersion [40]. Our studies identify that loss of *adcy5* interferes with melanosome dispersion in zebrafish embryos, juveniles, and adults. *adcy5* and *adcy3a* double mutations cause more severe defects in melanosome dispersion, presenting a punctate distribution of melanosomes in melanocytes. Zebrafish mlpha (melanophilin) mutation also displayed melanosome aggregation phenotypes similar to *adcy5* and *adcy3a* double mutations. However, the elevation of cAMP levels does not affect melanosome dispersion in *mlpha* mutant melanocytes. Zebrafish Mlpha acts downstream of cAMP and regulates dynein-dependent melanosome movements through an actin-independent mechanism [18]. In *adcy3a^-/^^-^*;*adcy5^-/^^-^* mutants, activating PKA signaling by the addition of cAMP analogue induces a significant dispersion of punctate melanosomes during embryonic and adult stages, indicating that the formation of punctate melanosomes is mainly due to their dispersion deficiency caused by cAMP-dependent PKA pathway inactivation. Consistent with this, the activation of kinesin-1 or the inhibition of cytoplasmic dynein-1, two potential effectors downstream of PKA, ameliorates high aggregation of melanosomes in *adcy3a^-/^^-^*;*adcy5^-/^^-^* double mutants, which further supports a role of Adcy3a and Adcy5 in regulating melanosome transport.

Mitf is a master regulator of melanocyte differentiation and melanogenesis, directly regulating melanocyte-specific genes such as *tyr, trp-1,* and *dct*. Zebrafish *nacr*e (mitfa) mutants completely lack melanocytes throughout development, leading to the absence of adult melanocyte stripes [4]. *adcy3a^-/^^-^*;*adcy5^-/^^-^* double mutants, similar to *nacr*e mutants, also fail to establish adult black stripes, but the total number of melanocytes including pigmented and unpigmented melanocytes appears to be normal, which is mainly due to reduced melanin production in *adcy3a^-/^^-^*;*adcy5^-/^^-^* double mutants by inhibiting the expression of *mitfa* and melanogenic enzymes, such as *tyr, trp-1,* and *dct*. In addition, *adcy3a^-/^^-^*;*adcy5^-/^^-^* double mutants have no effects on melanocyte specification and differentiation during the embryonic stage due to the normal expression of *sox10*, *mitfa, tyr,* and *dct*, exhibiting normal development of melanocytes during embryonic and metamorphic stages except for abnormal melanosome dispersion. However, several other zebrafish mutants that affect adult pigment stripe patterns, such as *sparse* (kita), *panthe*r (fms), and *rose* (ednrb1), lack either larval melanocytes or metamorphic melanocytes, leading to developing around half the normal complements of adult body stripe melanocytes [41].

MSCs are responsible for the morphogenesis and homeostasis of the adult zebrafish melanocyte pattern [28,29]. Defects in MSCs establishment in many zebrafish pigment mutants cause decreased melanocytes in the juvenile or adult stage [24,27,41]. Zebrafish *picasso* (erbb3b) mutants display the normal number of embryonic melanocytes but fail to develop normal metamorphic and adult melanocyte patterning due to the deficiencies in early neural crest cell migration and embryonic MSCs generation [23]. The *picasso* mutation also blocks melanocyte regeneration following embryonic melanocyte ablation by MoTP [28]. In contrast, *adcy3a* and *adcy5* double mutations have no effects on larval melanocyte regeneration, indicating that MSCs are well-established during embryonic stages. We noticed that *adcy3a* and *adcy5* double mutations have different effects on melanocyte maturation (melanocyte melanization) during embryonic and adult stages. There are two possible reasons for this phenomenon: (1) embryonic and adult melanocytes have different genetic origins; embryonic melanocytes develop directly from the neural crest, whereas adult melanocytes derive mainly from DRG-associated MSCs. (2) The maturation of melanocytes during the embryonic stage is regulated by the crosstalk between cAMP signaling and many other pathways [42], while this process may be mainly dependent on cAMP signaling in adulthood.

In mammals, there are nine isoforms of membrane-bound Adcys, categorized into four groups. Adcy5 and Adcy6 belong to Group III and share similar regulatory mechanisms, which are sensitive to inhibition by Ca2^+^. Although Adcy5 and Adcy6 are closely related isoforms, our studies demonstrate that only *adcy5* mutation causes abnormal pigmentation, but not the *adcy6a* mutation or *adcy6a* and *adcy6b* double mutations. Many studies have revealed that GPCRs display ADCY-specific coupling and mediate cAMP signaling compartmentation by organizing localized PKA signaling in various cellular processes [43,44,45]. Therefore, we believe that hormone receptors associated with the pigmentation process, such as Mc1r, might selectively couple to Adcy5 to stimulate cAMP generation in subcellular locations and guide skin pigmentation. As well as Adcy5, we identify that Adcy3a is also involved in the Adcy5-dominated skin pigmentation in zebrafish. Although *adcy3a* mutation does affect melanin patterns, its deletion further aggravated the pigmentation phenotype of *adcy5^-^^/^^-^* mutants, failing to establish melanin stripes in adult zebrafish. Since neither *adcy3^-/-^* nor *adcy5^-/-^* null mutant mice develop skin and hair pigmentation abnormalities, future studies will be required to elucidate the conservative mechanism of the specific action of Adcy3 and Adcy5 together across species in regulating skin and hair pigmentation.

## 4. Materials and Methods

### 4.1. Fish Stock and Embryo Culture

The wild-type zebrafish and Adcy isoforms mutants were raised under standard husbandry procedures. The Institutional Animal Care and Use Committee at East China Normal University advise animal care and research.

### 4.2. Generation of Zebrafish adcy3a^-/-^, adcy5^-/-^, and adcy6a^-/-^ Single Mutants, and adcy3a^-/-^;adcy3b^-/-^ and adcy6a^-/-^;adcy6b^-/-^ Double Mutants

The *adcy3a, adcy5,* and *adcy6a* single mutants, and *adcy3a^-/^^-^*;*adcy3b^-/^^-^* and *adcy6a^-/^^-^*;*adcy6b^-/^^-^* double mutants were generated using CRISPR/Cas9 technology. The sgRNA target site of *adcy3a, adcy3b,* and *adcy5* was in exon 6, *adcy6a* in exon 2, and *adcy6b* in exon 8, respectively. The corresponding sequences are as follows: *adcy3a* gRNA, 5′-CAACAAGATGGAGGCCGG -3′; *adcy5* gRNA, 5′-CTTGGTGAGGGAAGTCAC-3′; *adcy6*a gRNA, 5′-TGAGCTGCCTGCGGGACG-3′; *adcy3b* gRNA, 5′-TGGGCCTGTCTATGGTGG-3′; *adcy6b* 5′- AGGGTCTGATGCCTCGGT-3′. All gRNAs were synthesized using MAXIscript T7 Kit (Ambion). sgRNA (30 ng/μL) and Cas9 protein (New England Biolabs) were co-injected into zebrafish embryos at the one-cell stage. For *adcy3a* and *adcy5* double-gene knockout, the homozygous *adcy5* mutants and homozygous *adcy3a* mutants were outcrossed to obtain *adcy3a^-/^^-^*;*adcy5^-/^^-^* double mutants. The *adcy3a^-/^^-^*;*adcy3b^-/^^-^* and *adcy6a^-/^^-^*;*adcy6b^-/^^-^* mutants were generated by knocking out *adcy3b* in the *adcy3a^-/^^-^* mutants or *adcy6b^-/^^-^* in the *adcy6a^-/^^-^* mutants. For the rescue experiments, capped mRNA of *adcy5* and *adcy3a* was synthesized using the mMESSAGE mMACHINE Kit (Ambion, Thermo Fisher Scientific, Waltham, MA, USA). A total of 150 pg of *adcy5* mRNA and 150 pg of *adcy3a* mRNA were co-injected into *adcy3a^-/^^-^*;*adcy5^-/^^-^* double mutants at the one-cell stage.

### 4.3. Injection of Antisense Morpholino OLIGONUCLEOTIDES

Anti-sense morpholinos targeting the *adcy6a* start codon (*adcy6a-*MO: 5′-AAGCCGTTGATCCAGGACATTCTGT-3′), *dync1h1* start codon *(dync1h1*-MO: 5′-CGCCGCTGTCAGACATTTCCTACAC-3′) [37,38,39], and a standard control MO (control-MO: 5′-CCTCTTACCTCAGTTACAATTTATA-3′) was designed and synthesized from GeneTools. Morpholinos were dissolved in double distilled water at a concentration of 1 mM (8.33 mg/mL). Each embryo was injected in a dose of 1–2 ng.

### 4.4. Whole Mount In Situ Hybridization (WISH)

WISH was performed as described previously [46]. Digoxigenin-labeled RNA probes for *sox10*, *mitfa*, *tyr*, and *dct* were transcribed in vitro using the MAXIscript SP6/T7 Transcription Kit (AM1322, Invitrogen). In situ hybridization signals were detected using BM-Purple (11442074001, Roche).

### 4.5. RNA Isolation and Real-Time PCR Analysis

Total RNAs were extracted from wild-type embryos and *adcy3a^-/^^-^*;*adcy5^-/^^-^* mutant embryos, and the skin of adult wild-type males and *adcy3a^-/^^-^*;*adcy5^-/^^-^* mutant males using TRIZOL (Invitrogen), respectively. All males were about 6 months old. A total of 1 μg RNA was reverse transcribed into cDNA using PrimeScript™ II 1st strand cDNA Synthesis Kit (Takara). Real-time PCR was performed with SYBR Premix Ex Taq II (Takara) using Roche LightCycler^®^ 480 II PCR system. The relative expression levels of target genes were normalized to β-actin and quantified by the 2^−ΔΔCT^ method. For each biological sample, 3 biological replicates and 3 technical replicates were used. The following gene-specific primers were used for real-time PCR: *β-actin* forward 5′-GTTGACAACGG-CTCCGGTAT-3′ and reverse 5′-TTCTGTCCCATGCCAACCAT-3′; *mitfa* forward 5′-CCCTTCTCACCCTCAACTG-3′ and reverse 5′-GAACCCAAGAATGTCATCACT-3′; *dct* forward 5′-CCACAGTTCAGCAACATCTC-3′ and reverse 5′-ACCTGTCAGTTTCTG-TAGTTCTC-3′; *tyr* forward 5′-GACTTTAACTTCACCATCCC-3′ and reverse 5′-GAGAACA GATCACCTTCCA-3′; *trp1b* forward 5′-TGTGAGAGTGTGGAGGAG-3′ and reverse 5′-CTGTGGTTCTGGTAATCTTTGA-3′; *kita* forward 5′-CTATGTTGTCA-AAGGCAATGCT-3′ and reverse 5′-CCAGACGTCACTCTCA AAGGT-3′; *erbb*3b forward 5′- ACCTTGTGGTGAGGC CTGCTC-3′ and reverse 5′-CGCAAACCCAACCTGCAACC-3′.

### 4.6. Pigmentation Coverage Measurement and Melanocytes Count

We quantified the pigmentation coverage and melanocyte numbers with a defined region of interest (ROI) on the larval head. The ROI was restricted to the range from the middle of the eyes to the posterior of the hindbrain. The images of every larval head were acquired using a Leica M205FA stereomicroscope equipped with a Leica DFC7000T camera and submitted to ImageJ software. Image processing steps are as follows: (1) Selected images are opened in ImageJ. (2) the “straight line” tool is selected and then a straight line is drawn along the scale bar of the images. (3) Go to the “Set Scale” option in the Analyze menu and enter the length of the drawn line in the “Known distance” field in the pop-up dialog box, and then enter the unit in the “Unit of length” field. (4) Select the “freehand selections” tool and draw the freehand line to form a shape that closely surrounds the area portion of all melanocytes in the ROI. (5) Apply the “measure” option in the Analyze menu to obtain the statistic pigmentation area.

After quantifying the pigmentation coverage, zebrafish larvae were immersed in a 15 uM Norepinephrine bitartrate salt solution for 15 min to contract melanosomes, which facilitates the manual counting of melanocytes. Each melanocyte is visible after treatment and the number of melanocytes in the ROI can be counted directly.

### 4.7. Melanosome Dispersion Experiments

α-MSH (Target Mol), db-cAMP (Target Mol), kinesore (GLPBio), and ciliobrevin D (Target Mol) were dissolved in DMSO for a 50 mM stock solution. A brief description of each test procedure is as follows. 10-day-old *adcy3a^-/^^-^*;*adcy5^-/^^-^* mutant larvae were imaged using a Leica M205FA stereomicroscope, and then incubated in DMSO (vehicle control), 1 mM α-MSH, and 5 mM db-cAMP for 30 min, respectively, providing sufficient time for melanosome dispersal. Subsequently, these treated embryos were subjected to image again. In adult zebrafish, a piece of the fin tail was detached from *adcy3a^-/^^-^*;*adcy5^-/^^-^* double mutants with a scalpel as previously reported methods [40], washed briefly in physiological saline solution, and then incubated in DMSO, 2 mM α-MSH and 10 mM db-cAMP for 30 min, respectively. The fin tail was imaged before and after treatment with these small molecule compounds. To assess the effects of activation of kinesin-1 and inhibition of cytoplasmic dynein-1 on melanosome dispersion, WT embryos, and *adcy3a^-/^^-^*;*adcy5^-/^^-^* mutant embryos were treated with 50 μM kinesore and 20 μM ciliobrevin D from 24hpf to 96hpf, respectively, then were subjected to the image. The areas of melanosome dispersion were analyzed using ImageJ software.

### 4.8. MoTP-Induced Melanocyte Ablation

As previously reported [31], zebrafish embryos were incubated in 200 μM (2-morpholinobutyl)-4-thiophenol) (MoTP) (GLPBIO) from 1 dpf to 3 dpf to ablate differentiated melanocytes and then counted for the regenerated melanocytes at 6 dpf following the washout of 4-MoTP at 3 dpf.

## 5. Conclusions

Melanin pigmentation not only determines the colors of hair and skin but also protects skin from ultraviolet radiation. Cyclic adenosine monophosphate (cAMP) is the central moderator responsible for responding to various signals, including hormones, neurotransmitters, and light, to regulate the pigmentation of hair and skin. In mammals, the ADCY family consists of ten members accountable for producing cAMP. However, which ADCY isoforms can be especially involved in melanocyte development, melanin synthesis, and melanosome transport in vivo remains unknown. Here, we generated a series of single and double mutants of Adcy isoforms using zebrafish as a model. Through the analyses of these mutants, we show that *adcy3a* and *adcy5* double mutations cause high aggregated melanosomes in melanocytes during zebrafish embryonic and juvenile stages. Surprisingly, the loss of *adcy3a* and *adcy5* results in the complete disappearance of melanocyte stripes in adult zebrafish, presenting with a few punctate pigmented melanocytes and many unpigmented melanocytes, which makes the appearance of the skin more transparent. In conclusion, we demonstrate that Adcy3a and Adcy5 are required for melanin synthesis and melanosome dispersion, but not melanocyte differentiation.

## Figures and Tables

**Figure 1 ijms-23-14182-f001:**
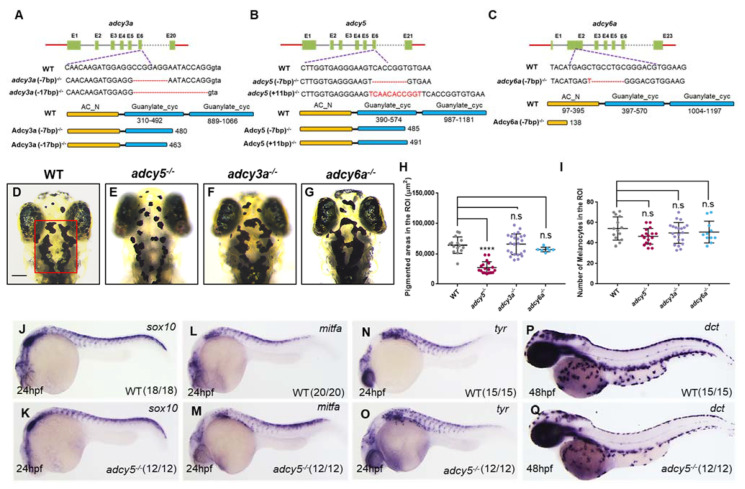
*adcy5* mutation affects melanosome dispersion during embryonic stages. (**A**–**C**) The sgRNA target is designed in the sixth exon of *adcy3a* (**A**) and *adcy5* (**B**), and the second exon of *adcy6a* (**C**). Schematic representations of the nucleotide sequences show a 7-bp and a 17-bp deletion in *adcy3a^-/^^-^* (**A**), a 7-bp deletion and an 11-bp insertion in *adcy5^-/^^-^* (**B**), and a 7-bp deletion in *adcy6a^-/^^-^* (**C**). Red dashes indicate the deleted bases; the inserted bases are marked in red. Predicted domain structure of Adcy3a, Adcy5, and Adcy6a from wild-type and mutant amino acid sequences. (**D**–**G**) *adcy3a^-/^^-^* (**F**) and *adcy6a^-/^^-^* (**G**) single mutants show normal melanosome dispersion as seen in WT embryos (**D**) under dark conditions at 4dpf, whereas *adcy5^-/^^-^* single mutants (**E**) exhibit aggregated melanosomes. The red box in (**D**) indicates a defined region of interest (ROI) to quantify pigmentation coverage and melanocyte numbers. The ROI was restricted to the range from the middle of the eyes to the posterior of the hindbrain. Scale bar = 100 μm in (**D**–**G**). (**H**) Pigmentation coverage in the ROI is significantly decreased in *adcy5^-/^^-^* mutants compared to WT embryos (Student’s *t*-test, **** *p* < 0.0001, n.s., not significant, *p* > 0.05). (**I**) Statistical analyses show that the number of melanocytes in *adcy3a^-/^^-^*, *adcy5^-/^^-^*, and *adcy6a^-/^^-^* is comparable with that of WT embryos (Student’s *t*-test, n.s., not significant, *p* > 0.05). (**J**–**Q**) WISH analyses show the expression levels of *sox10*, *mitfa*, *tyr*, and *dct* in *adcy5^-/^^-^* mutants are comparable with those in WT embryos.

**Figure 2 ijms-23-14182-f002:**
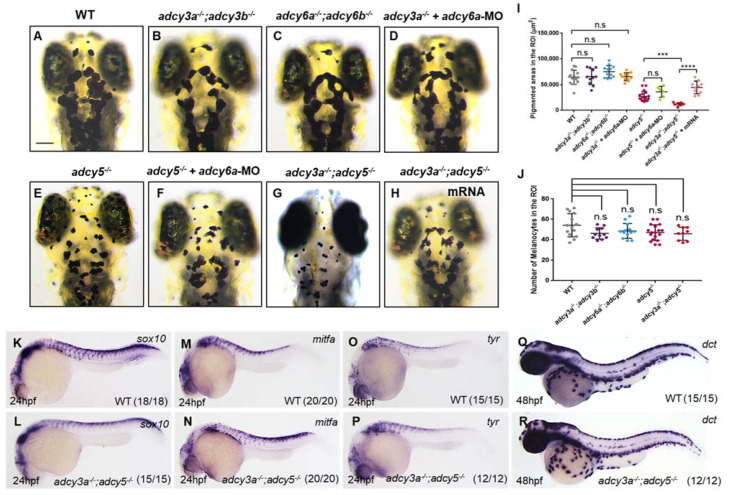
*adcy3a^-/^^-^* and *adcy5^-/^^-^* double mutants show high melanosome aggregation. (**A**–**H**) *adcy3a^-/^^-^*;*adcy3b^-/^^-^* (**B**) and *adcy6a^-/^^-^*;*adcy6b^-/^^-^* (**C**) double mutants as well as *adcy6a*-MO-injected *adcy3a^-/^^-^* mutants (**D**) display normal distribution of melanosomes as that of WT embryos (**A**) under dark conditions at 4dpf. *adcy6a*-MO injected *adcy5^-/^^-^* mutants (**F**) exhibit comparable pigmentation as *adcy5^-/^^-^* (**E**). *adcy3a^-/^^-^*;*adcy5^-/^^-^* double mutants (**G**) develop a higher degree of aggregation of melanosomes than *adcy5^-/^^-^*, while co-injection of *adcy3a* and *adcy5* mRNA can significantly rescue the defects in melanosome aggregation in *adcy3a^-/^^-^*;*adcy5^-/^^-^* double mutants (**H**). Scale bar = 100 μm in (**A**–**H**). (**I**) Pigmentation coverage shows no difference among wild-type embryos, *adcy3a^-/^^-^*;*adcy3b^-/^^-^*, *adcy6a^-/^^-^*;*adcy6b^-/^^-^,* and *adcy3a^-/^^-^* mutants injected with *adcy6a*-MO, as well as between *adcy5^-/^^-^* and *adcy5^-/^^-^* mutants injected with *adcy6a*-MO. *adcy3a^-/^^-^*;*adcy5^-/^^-^* double mutants have significantly reduced pigmentation areas compared to *adcy5^-/^^-^,* which can be rescued by co-injection of *adcy3a* and *adcy5* mRNA (Student’s *t*-test, *** *p* < 0.001, **** *p* < 0.0001, n.s., not significant, *p* > 0.05). (**J**) Statistical analyses show that the number of melanocytes in *adcy5^-/^^-^* single mutants, and a*dcy3a^-/^^-^*;*adcy3b^-/^^-^*, *adcy6a^-/^^-^*;*adcy6b^-/^^-^* and *adcy3a^-/^^-^*;*adcy5^-/^^-^* double mutants is comparable with that of WT embryos (Student’s *t*-test, n.s., not significant, *p* > 0.05). (**K**–**R**) WISH analyses showed the expression levels of *sox10*, *mitfa*, *tyr*, and *dct* are normal in *adcy5*^-/-^;*adcy3a^-/^^-^* compared to WT embryos.

**Figure 3 ijms-23-14182-f003:**
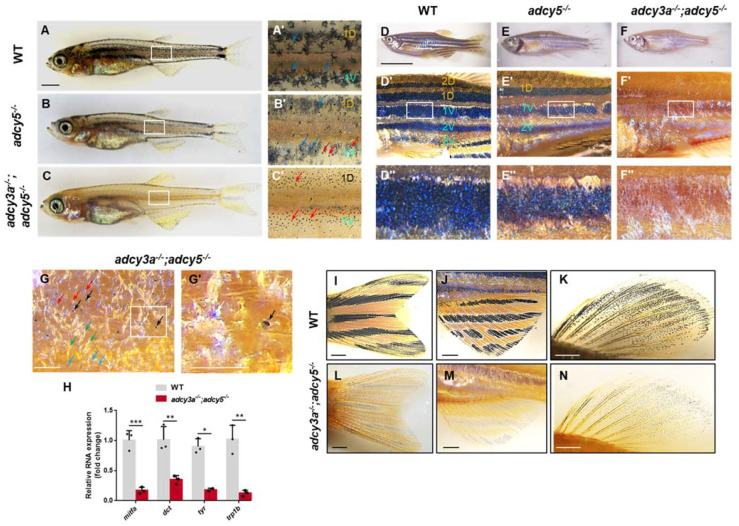
*adcy3a and adcy5* are required for adult melanin patterning. (**A**–**C**) WT zebrafish develop metamorphic melanocyte stripes around 30 days, including 1D and 1V, along with corolla-like patterns of melanosomes in most melanocytes (**A**). *adcy5^-/^^-^* mutants establish normal metamorphic stripes, but some melanocytes present punctate distribution of melanosomes (**B**). *adcy3a^-/^^-^*;*adcy5^-/^^-^* double mutants are unable to form corolla distribution of melanosomes and instead punctate melanosomes (**C**). (**A′**–**C′**) are the magnified images of boxes in (**A**–**C**) (the red arrows indicate punctate melanocytes; the blue arrows indicate corolla melanocytes). Scale bar = 1000 μm in (**A**–**C**). (**D**–**F**) Adult wild-type zebrafish form five melanocyte stripes, including three ventral black stripes 1V, 2V, and 3V, and two dorsal stripes 1D and 2D along the dorsoventral axis (**D**). *adcy5^-/^^-^* mutants (**E**) show three melanocyte stripes, including black stripes 1D, 1V, and 2V, with a lower density of melanocytes, compare to WT zebrafish (**F**). *adcy3a^-/^^-^*;*adcy5^-/^^-^* double mutants fail to establish normal melanocyte stripes, along with a few punctate melanocytes (**F**). (**D′**–**F′**) are the partially magnified images of (**D**–**F**), and (**D″**–**F″**) are the magnified images of boxes in (**D′**–**F′**). 1D: dorsal primary black stripe; 1V: ventral primary black stripe; 2D: dorsal secondary black stripe; 2V: ventral secondary black stripe; 3V: the third ventral black stripe. Scale bar = 10 mm in (**D**–**F**). (**G**) Adult *adcy3a^-/^^-^*;*adcy5^-/^^-^* double mutants develop a few pigmented melanocytes, along with many unpigmented melanocytes (the red arrows indicate unpigmented melanocytes; the black arrows indicate pigmented melanocytes; the blue arrows indicate xanthophores; the green arrows indicate iridophores). (**G′**) is the partially magnified images of boxes in (**G**). Scale bar = 100 μm. (**H**) qPCR analyses of *mitfa, dct, tyr,* and *trp1b* expression in the adult skin from WT zebrafish or *adcy3a^-/^^-^*;*adcy5^-/^^-^* mutants (Student’s *t*-test, * *p* < 0.05, ** *p* < 0.01, *** *p* < 0.001). (**I**–**N**) The caudal, anal, and dorsal fins in *adcy3a^-/^^-^*;*adcy5^-/^^-^* double mutants show highly aggregated melanosomes compared with those in WT zebrafish. Scale bar = 1000 μm.

**Figure 4 ijms-23-14182-f004:**
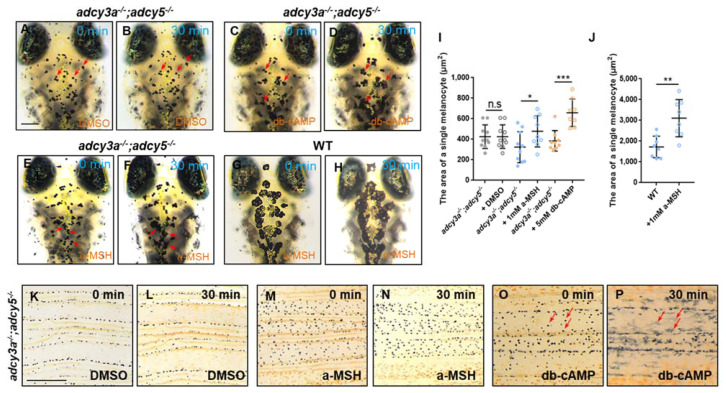
Activation of PKA ameliorates melanosome aggregation in *adcy3a^-/^^-^*;*adcy5^-/^^-^* double mutants. (**A**–**H**) 10-day-old *adcy3a^-/^^-^*;*adcy5^-/^^-^* double mutants have no response to DMSO treatment (**A**,**B**), but show significant dispersion of melanosomes when exposed to db-cAMP for 30 min (**C**,**D**). The melanosomes in a few melanocytes show mild dispersion when exposed *adcy3a^-/^^-^*;*adcy5^-/^^-^* double mutants to a-MSH for 30 min (**E**,**F**), while they dramatically disperse in almost all melanocytes in WT larvae (**G**,**H**). The red arrows indicate the same melanocytes from the same embryo before and after treatment with chemical compounds. Scale bar = 250 μm in (**A**–**H**). (**I**,**J**) Statistical analyses of the pigmented area of single melanocyte in *adcy3a^-/^^-^*;*adcy5^-/^^-^* double mutants (**I**) and WT embryos (**J**) before and after treatment with DMSO, a-MSH, and db-cAMP, respectively (Student’s *t*-test n.s., not significant, *p* > 0.05, * *p* < 0.05. ** *p* < 0.01. *** *p* < 0.001). (**K**–**P**) No melanosome dispersion is detected in DMSO (**K**,**L**) or a-MSH-treated caudal fins (**M**,**N**) derive from *adcy3a^-/^^-^*;*adcy5^-/^^-^* double mutants, but almost all melanosomes disperse in db-cAMP-treated mutant caudal fins (**O**,**P**). The red arrows indicate some dramatically dispersed melanosomes from the same caudal fin before and after treatment with db-cAMP. Scale bar = 250 μm in (**K**–**P**).

**Figure 5 ijms-23-14182-f005:**
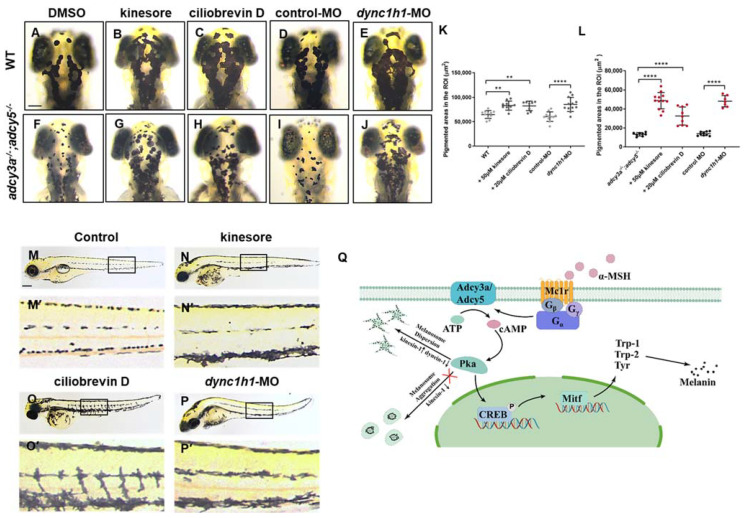
Activation of kinesin-1 or Inhibition of cytoplasmic dynein-1 rescues the phenotype of high melanosome aggregation in *adcy3a^-/^^-^*;*adcy5^-/^^-^* double mutants. (**A**–**C**) WT embryos treated with either kinesore or ciliobrevin D from 24 hpf to 96 hpf show high melanosome dispersion when compared with DMSO controls. (**D**,**E**) *dync1h1*-MO-injected WT embryos display significant melanosome dispersion compared with control-MO-injected embryos. (**F**–**H**) *adcy3a^-/-^*;*adcy5^-/-^* double mutants show significant dispersion of melanosomes when treated with either kinesore or ciliobrevin D from 24 hpf to 96 hpf compared with DMSO-treated groups. (**I**,**J**) *adcy3a^-/-^*;*adcy5^-/-^* double mutants injected with *dync1h1*-MO exhibit significant dispersion of melanosomes in comparison with controls. (**K**,**L**) Statistical analysis of pigmentation coverage in WT embryos (**K**) and *adcy3a^-/-^*;*adcy5^-/-^* double mutants (**L**) treated with DMSO, kinesore, and ciliobrevin D, respectively, as well as either injected with *dync1h1*-MO or control-MO (Student’s *t*-test, ** *p* < 0.01. **** *p* < 0.0001). (**M**–**O**) The dispersion of melanosomes throughout the body is detected in kinesore (**N**,**N′**) or ciliobrevin D-treated *adcy3a^-/-^*;*adcy5^-/-^* double mutants (**O**,**O′**), compared with DMSO-treated groups (**M**,**M′**). (**M′**–**O′**) are the partially magnified images of boxes in (**M**–**O**). (**P**) *dync1h1*-MO injected *adcy3a^-/-^*;*adcy5^-/-^* double mutants display melanosome dispersion throughout the body. (**P′**) is the partially magnified images of boxes in (**P**). (**Q**) A working model for Adcy3a and Adcy5 signaling cascades involving melanosome dispersion and melanin synthesis: α-MSH binds to Mc1r and activates Adcy3a and Adcy5, leading to intracellular cAMP increase; the increased cAMP stimulates Pka to guide melanosome dispersion by activating kinesin-1 and suppressing cytoplasmic dynein-1, and promotes melanin synthesis through increasing the expression of *mitf* and its downstream target genes *tyr*, *trp-1*, and *trp-2*. Scale bar = 100 μm in (**A**–**J**). Scale bar = 250 μm in (**M**–**P**).

## Data Availability

The data presented in this study are available in Appendix A.

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
