# Peer review of "Requirement of Zebrafish Adcy3a and Adcy5 in Melanosome Dispersion and Melanocyte Stripe Formation"

_ijms, 2022, doi:10.3390/ijms232214182_

Round 1
Reviewer 1 Report
This article findings reveal roles of Adcy3a and Adcy5 in regulating pigmentation in zebrafish model. Over all the manuscript is interesting and bring a novel tool. Following are some suggestions:
- The flow of text and figures need to be organized.
- The authors should compare other studies done using zebrafish model on pigmentation studies.
Author Response
Reviewer 1
Point 1: The flow of text and figures need to be organized.
Response: Thank the reviewer for the suggestion. We reorganized the text and enhanced its readability and logic according to the figure arrangement. We rearranged some images, such as Figure 3, making them more reasonable and beautiful.
Point 2: The authors should compare other studies done using zebrafish model on pigmentation studies.
Response: We understand the reviewer’s comments and have revised the manuscript. We compared adcy3a-/-adcy5-/- double mutants with other pigment mutants in zebrafish and clarified their respective phenotypic characteristics in the Discussion section.
Reviewer 2 Report
Zhang et al. describe the important role of adenylate cyclase 3a and 5a in melanosome dispersion and melanocyte stripe formation. Using CRISPR/Cas9 approach, the authors generated zebrafish mutants for various isoforms of adcy. The authors show that adcy3a and adcy5 double mutant has stronger phenotype compared to other mutants. Adcy3a and adcy5 double mutants cause high aggregated melanosomes in melanocytes during zebrafish embryonic and juvenile stages. Surprisingly, loss of adcy3a and adcy5 results in the complete disappearance of melanocyte stripes in adult zebrafish, presenting with a few punctate pigmented melanocytes and many unpigmented melanocytes, which makes the appearance of the skin more transparent. The authors demonstrate that Adcy3a and Adcy5 are required for melanin synthesis and melanosome dispersion, but not melanocyte differentiation. The objective of the study was clearly defined, and experiments were well planned, accordingly. Overall, I am glad to review this manuscript and I recommend this article to be published in IJMS, with one important revision.
Minor Revision:
Throughout the manuscript, the authors have measured the pigmentation coverage and counted the melanocytes. However, the method is not clearly explained, especially for the general audience. I advise the authors to add a detailed methodology for this experiment, if possible, add a representative image.
Author Response
Reviewer 2
Throughout the manuscript, the authors have measured the pigmentation coverage and counted the melanocytes. However, the method is not clearly explained, especially for the general audience. I advise the authors to add a detailed methodology for this experiment, if possible, add a representative image.
Response: We understand the reviewer’s comments and have revised the manuscript. We add a detailed methodology for quantifying pigmentation coverage measurement and melanocyte count in the “Materials and Methods” section. It will make readers understand the relevant research content more clearly. We tried to describe it by schematic drawings, but this doesn't make it any better.